



# Carbonate pump feedbacks on alkalinity and the carbon cycle in the 21[st] century and beyond

Alban Planchat[1], Laurent Bopp[1], Lester Kwiatkowski[2], and Olivier Torres[1]

[1]LMD-IPSL, CNRS, Ecole Normale Supérieure/PSL Res. Univ, Ecole Polytechnique, Sorbonne Université, Paris, 75005, France
[2]LOCEAN Laboratory, Sorbonne Université-CNRS-IRD-MNHN, Paris, 75005, France

**Correspondence:** Alban Planchat (alban.planchat@lmd.ipsl.fr)

**Abstract.** Ocean acidification is likely to impact all stages of the ocean carbonate pump, i.e. the production, export, dissolution and burial of biogenic $CaCO_3$. However, the associated feedbacks on anthropogenic carbon uptake and ocean acidification have received little attention. It has previously been shown that Earth system model (ESM) carbonate pump parameterizations can affect and drive biases in the representation of ocean alkalinity, which is critical to the uptake of atmospheric carbon and

provides buffering capacity towards associated acidification. In the sixth phase of the Coupled Model Intercomparison Project (CMIP6), we show divergent responses of $CaCO_3$ export at 100 m this century, with anomalies by 2100 ranging from -74 % to +23 % under a high-emissions scenario. The greatest export declines are projected by ESMs that consider pelagic $CaCO_3$ production to depend on the local calcite/aragonite saturation state. Despite the potential effects of other processes on alkalinity, there is a robust negative correlation between anomalies in $CaCO_3$ export and salinity-normalized surface alkalinity across the

CMIP6 ensemble. Motivated by this relationship and the uncertainty in $CaCO_3$ export projections across ESMs, we perform idealized simulations with an ocean biogeochemical model and confirm a limited impact of carbonate pump anomalies on twenty-first century ocean carbon uptake and acidification. However between 2100 and 2300, we highlight a potentially abrupt shift in the dissolution of $CaCO_3$ from deep to subsurface waters when the global scale mean calcite saturation state reaches about 1.23 at 500 m (likely when atmospheric $CO_2$ reaches 900 to 1100 ppm). During this shift, upper ocean acidification due

to anthropogenic carbon uptake induces deep ocean acidification driven by a substantial reduction in $CaCO_3$ deep dissolution following its decreased export at depth. Although the effect of a diminished carbonate pump on global ocean carbon uptake and surface ocean acidification remains limited until 2300, it can have a large impact on regional air-sea carbon fluxes, particularly in the Southern Ocean.

## 1 Introduction

Plankton are the basis of the oceanic food chain. They provide both structure and function to marine ecosystems. In addition, by producing organic matter and calcium carbonate – in the case of calcifying plankton – which is exported as particulates to the ocean interior, they act as a biological carbon pump, which is central to understanding the ocean carbon cycle (Hain et al., 2014). Each year, these organisms facilitate the export of about 10 PgC of organic matter and calcium carbonate from the surface ocean to the interior ocean (DeVries and Weber, 2017; Sulpis et al., 2021, with a range of uncertainty of about 4 to



16 Pg yr$^{-1}$). Thus, in the face of increasing atmospheric $CO_2$ concentration and climate change resulting from human activities, it is crucial to understand the response of these organisms to emerging environmental stressors (e.g. acidification, increased temperature, deoxygenation and reduced nutrient availability, Kwiatkowski et al., 2020). A particular focus is accorded to the surface ocean where most of the feedbacks between marine biology and the global Earth system are expected to occur this century (Cooley et al., 2022).

The soft-tissue and carbonate pumps, which together make up the biological pump, have opposing effects on the surface carbon cycle by modifying dissolved inorganic carbon (DIC) and total alkalinity (Alk) differently (e.g. Zeebe et al., 2001; Sarmiento and Gruber, 2006; Hain et al., 2014). Organic matter production results in $CO_2$ uptake and a basification of the surface ocean. In contrast, the production of calcium carbonate ($CaCO_3$) causes a release of $CO_2$ and an acidification of the surface ocean. Hence, the carbonate pump effectively exports carbon from the surface towards the ocean interior while

promoting a degassing of $CO_2$ towards the atmosphere, which explains why it is called "carbonate counter-pump". While the soft-tissue pump is key in driving the vertical DIC gradient, the carbonate pump, about an order of magnitude smaller in terms of carbon export at 100 m, is important for driving the vertical Alk gradient (Sarmiento and Gruber, 2006; Planchat et al., 2023). One should therefore consider the two pumps separately, particularly with respect to the export of particulate organic carbon (POC) in the case of the soft-tissue pump, and of particulate inorganic carbon (PIC) in the case of the carbonate pump.

The rain ratio – defined as the ratio between the exports of PIC and POC – (e.g. Zeebe et al., 2001) has long been of interest to modellers in their quest for implicit parameterization of the carbonate pump based on the representation of the soft-tissue pump (Planchat et al., 2023). But the rain ratio is also central in assessing the effects of the biological pump on the surface carbon cycle. For example, it is possible to express the Revelle factor, which relates relative anomalies of DIC and $p$CO$_2$, as a function of the rain ratio (Zeebe et al., 2001).

In the framework of the Coupled Model Intercomparison Projects (CMIPs), a number of Earth system model (ESM) comparisons have analysed soft-tissue pump projections, typically focusing on net primary production (NPP) and POC export (Bopp et al., 2013; Laufkötter et al., 2015, 2016; Fu et al., 2016; Kwiatkowski et al., 2020; Tagliabue et al., 2021; Henson et al., 2022; Wilson et al., 2022, for CMIP5 and CMIP6). However, no comparison studies exist to date on carbonate pump projections, and especially PIC export. Yet, biological studies carried out on calcifiers raise questions regarding their response to climate change and acidification (Gattuso and Hansson, 2011). Although laboratory and field-data meta-analyses generally

support an expected decrease in calcification due to ocean acidification (e.g. Kroeker et al., 2013; Meyer and Riebesell, 2015; Seifert et al., 2020) – especially for planktonic calcifiers, calcifying algae and corals (Leung et al., 2022) – uncertainties are high due to potential decoupling between growth and calcification in response to environmental stressors (e.g. light and nutrient availability, as well as carbonate chemistry, Zondervan et al., 2001; Seifert et al., 2022). In addition, biological studies often focus on coccolithophores, and in particular *Emiliane huxleyi*, which may not be representative of wider pelagic calcifiers

(Ridgwell et al., 2007), which exhibit diverse responses to environmental change (e.g. Kroeker et al., 2013). All these considerations make it difficult to constrain model parameterizations with confidence. Modelling studies carried out to assess potential feedbacks of the carbonate pump in response to climate change and acidification have thus shown diverging results essentially depending on the $CaCO_3$ production parameterization. While a climate-driven increase in the future growth rates of certain





calcifying species has been projected (Schmittner et al., 2008), a $CO_2$-driven decrease in calcification in response to ocean acidification has also been projected (Gehlen et al., 2007; Ridgwell et al., 2007; Ilyina et al., 2009; Hofmann and Schellnhuber, 2009; Gangstø et al., 2011; Pinsonneault et al., 2012). The former induces a positive climate feedback as opposed to the latter, emphasizing the potential importance of considering these effects with an explicit representation of calcifiers (Krumhardt et al., 2019). Despite this, all current ESMs implicitly model $CaCO_3$ production based on POC production, and rarely with a

saturation-state dependency (Planchat et al., 2023). Models also typically consider calcite and not aragonite production, which may induce delays in the response of the carbonate pump to acidification, as aragonite is less stable than calcite. Similarly, models may underestimate carbonate pump feedbacks by not representing benthic calcifiers, such as corals, which are likely to be particularly vulnerable to climate change (Bindoff et al., 2019).

   On paleoclimatic time scales, especially for the study of glacial-interglacial transitions, the oceanic $CaCO_3$ cycle is often

invoked to explain changes in the global carbon cycle, and in particular variability in the concentration of atmospheric $CO_2$ of about 80 to 100 ppm (Sigman and Boyle, 2000). Changes in the carbonate pump magnitude, in particular through shallow water coral reef surface availability, could partly drive large variations in the concentration of atmospheric $CO_2$ (e.g. Ridgwell et al., 2003). In addition, in face of a perturbation in the carbonate chemistry, the carbonate compensation feedback tends to restore the balance between river input of Alk and $CaCO_3$ burial through fluctuations of the lysocline depth – the upper limit

of the transition zone, where sinking and sedimentary $CaCO_3$ starts to substantially dissolve – at a time scale of about $10^4$ years (e.g. Broecker and Peng, 1987; Sigman and Boyle, 2000; Sarmiento and Gruber, 2006; Boudreau et al., 2018; Kurahashi-Nakamura et al., 2022). This mechanism alleviates an initial perturbation on atmospheric $CO_2$ from an external source to the ocean (negative feedback; e.g. for an imbalance in the terrestrial carbon cycle) and amplifies it when resulting from an internal ocean process (positive feedback; e.g. for a change in the organic matter or $CaCO_3$ production; Sarmiento and Gruber, 2006).

In this study, we are interested in how the future carbonate pump, as simulated in ESMs, responds to the anthropogenic perturbation of the carbon cycle, the speed and amplitude of which are greater than those of glacial-interglacial transitions. We also explore the feedbacks associated with this modification on the oceanic carbon cycle on multi-centennial time scales. To do so, we compare PIC export projections under high twenty-first century anthropogenic emissions in the CMIP6 model ensemble. Finally, to gain insight into the potential influence of PIC export biases and trends on projections of future ocean

carbon uptake and acidification, we perform sensitivity simulations using the NEMO-PISCES marine biogeochemical model.

## 2   Methodology

### 2.1   CMIP6 ESMs and outputs

We assess 15 ESMs from 12 different climate modelling centers (CCCma, CMCC, CNRM-CERFACS, CSIRO, IPSL, MIROC, MOHC, MPI-M, MRI, NCAR, NCC and NOAA-GFDL), which took part in the sixth phase of the Climate Model Intercompar-

ison Project (CMIP6, Eyring et al., 2016). All ESMs considered in this study represent the carbonate pump with the production of pelagic $CaCO_3$, its export in the form of PIC, as well as its dissolution and potential burial at depth. While we analysed only one ensemble member per ESM, we assessed two distinct ESMs for MPI-M since two different resolutions were available, as





well as for CCCma and NOAA-GFDL for which two distinct marine biogeochemical models were available (see Table A1). For each of the ESMs, we considered the piControl (pre-industrial control simulation), the Historical (covering the recent past, from 1850 to 2014) and the two extreme CMIP6 Shared Socio-economic Pathways (SSPs), SSP1-2.6 and SSP5-8.5. While our focus is on the SSP5-8.5 scenario, we use the SSP1-2.6 scenario, not available for MRI-ESM2-0 and GFDL-CM4, as a point of comparison. Each ESM is weighted in the calculation of CMIP6 statistical values (mean, standard deviation, quartiles, and linear regressions) such that each modeling group has the same total contribution.

The following variables were processed when available: (i) two-dimensional (2D) variables: sinking fluxes at 100 m of organic matter, calcite and aragonite (respectively, 'epc100', 'epcal100' and 'eparag100') and the $CO_2$ gas exchange between the ocean and the atmosphere ('fgCO$_2$'); three-dimensional (3D) variables: total alkalinity ('talk'), dissolved inorganic carbon ('dissic'), phosphate and silicate concentrations ('po4' and 'si'), salinity ('so') and the potential temperature ('thetao'). In order to facilitate the ESM intercomparison, we used the distance-weighted average remapping 'remapdis' and linear level interpolation with extrapolation 'intlevelx' of Climate Data Operator (CDO) to regrid the data on a regular 1°x1° grid with 33 depth levels, from 5 to 5500 m. The ocean carbonate system was computed with mocsy 2.0 (Orr and Epitalon, 2015) over the simulation period using annual (i) Alk, DIC, salinity and temperature from ESM outputs, (ii) phosphate and silicate from ESMs or from the gridded observational-based product of the Global Ocean Data Analysis Project (GLODAPv2.2020, Olsen et al., 2020) if at least one of the two was not simulated, and (iii) the seawater equilibrium constants recommended for best practices (Dickson et al., 2007; Orr and Epitalon, 2015). Quality control of ESM outputs led us to: (i) exclude grid cells with a salinity lower than 25 g kg$^{-1}$ in the analysis of processes including salinity data due to salinity normalization issues, and (ii) restrict the CNRM-ESM2-1 domain to exclude grid cells in the Japan Sea that exhibit highly anomalous PIC export.

## 2.2 NEMO-PISCES ocean biogeochemical model and sensitivity simulations

We used the marine biogeochemical model NEMO-PISCES to evaluate how a steady-state bias, or a change in the carbonate pump magnitude, can impact the ocean carbon cycle in the Anthropocene, suggested to start at the time of the Industrial Revolution (Crutzen and Stoermer, 2000). Although globally similar to the PISCES version described in Aumont et al. (2015), and used in IPSL-CM6A-LR, we made two changes: (i) the N-fixation parameterization was modified following Bopp et al. (2022), and (ii) the burial fraction of PIC was adjusted so that the global Alk inventory is conserved without the need of an Alk restoring scheme (see Appendix A2). We performed offline simulations on a tripolar ORCA grid with a nominal resolution of 2° and 30 vertical levels. While keeping pre-industrial ocean physics from an ESM used in CMIP5 by IPSL (IPSL-CM5A-LR, Dufresne et al., 2013), we completed 500-year runs (from 1800 to 2299) with the atmospheric $CO_2$ concentration following that of the Historical and then RCP-8.5 scenario (Representative Concentration Pathway). Although slightly less extreme than the SSP5-8.5 scenario, reaching 936 ppm in 2100 as opposed to 1135 ppm for the latter (see Appendix A3), this suited this analysis by validating some orders of magnitude, while using relatively coarse resolution simulations with low computational cost, as the CMIP6 scenarios were only available at higher resolution for the IPSL ESM. Two configurations with varying carbonate pump magnitudes were brought to steady-state with a 2500-yr spin-up, during which the burial fraction of PIC is free to evolve, prior to being fixed at the end of the spin-up (see Table A2). Then, we performed two sensitivity simulations



in parallel. First, we extended the two simulations brought to equilibrium according to the Historical and RCP-8.5 scenario (referred to 'standard' and 'carb_low' in the following) to assess the consequences of an equilibrium bias in the amplitude of the carbonate pump. Second, we performed two simulations for which we imposed a decrease or an increase (respectively referred to 'carb–' and 'carb+' in the following) of the carbonate pump as a function of the atmospheric $CO_2$ concentration in order to estimate the impact of a variation in the carbonate pump magnitude on the carbon cycle. In both simulations, the PIC production in the model was multiplied by -$\alpha_{carb}$ for carb– and +$\alpha_{carb}$ for carb+, with

$$\alpha_{carb} = 0.15 \cdot \frac{p\mathrm{CO}_2^{\mathrm{atm}} - 285}{936 - 285} \tag{1}$$

where 285 ppm is the atmospheric $CO_2$ concentration in 1850 and 936 ppm is the one in 2100 in RCP-8.5. A dependency on the atmospheric $CO_2$ concentration was preferred to a saturation state dependency to avoid any feedback from PIC production on the saturation state, which in turn influences PIC production itself. This also permitted control of the PIC production values reached in the sensitivity experiments so that they can be directly compared against one another and to the CMIP6 ensemble. Additional sensitivity simulations of the standard configuration were performed in support of the main analysis, taking into account only climate-driven effects (referred to 'standard_dyn'), or both climate-driven and $CO_2$-driven effects (referred to 'standard_dyn+atm'; see Table A2).

## 2.3 Technical processing

We used the normalization approach of dividing surface Alk and DIC values by the coincident salinity and multiplying this by a reference salinity value of 35 g kg$^{-1}$ (e.g. Sarmiento and Gruber, 2006; Fry et al., 2015) to remove the impact of freshwater fluxes (e.g. precipitation, evaporation, sea ice formation/melting and river discharge, Friis et al., 2003). Hereafter, salinity-normalized Alk and DIC are respectively referred to as sAlk and sDIC. We report a surface drift in sAlk for both CMCC-ESM2 and CNRM-ESM2-1. This is due to an Alk drift in CNRM-ESM2-1 and a salinity drift in CMCC-ESM2 (see Appendix A5). As a result, these two ESMs are excluded when reporting ESM values that may be affected by such a drift.

In the following, we refer to $CaCO_3$ without distinguishing between calcite and aragonite, unless explicit. This distinction is only relevant for GFDL-ESM4, which is the only ESM to represent both calcite and aragonite in CMIP6. In contrast, most groups simulate only calcite while ACCESS-ESM1-5 simulates a generic $CaCO_3$. Moreover, as calcite is the calcium carbonate mineral most commonly considered in ESMs, we use the calcite saturation state as a proxy for ocean acidification. We also define the rain ratio at 100 m as the ratio of the spatially integrated PIC and POC exports at 100 m.

Furthermore, we did not debias simulated carbonate chemistry of both ESMs and NEMO-PISCES runs in order to maintain consistency in the physico-chemistry experienced by marine biogeochemical processes within each model. We applied 11-year rolling means to time series, and when considering absolute values or anomalies, we averaged over 20-year periods to limit the influence of interannual variability which can be considerable for PIC export.





# 3    Results and discussion

## 3.1    Divergent PIC export projections in CMIP6 ESMs

Twenty-first century projections of PIC export are highly divergent in CMIP6 ESMs (from -74 % to +23 % in 2081-2100 for
SSP5-8.5; Fig. 1a), while projections of POC export are more consistent (-21 % to +3 %; see Fig. B3b). The divergent trend in
the rain ratio (from -69 % to +30 %; Fig. 1b) is therefore predominantly controlled by changes in PIC export. Nonetheless, the
maximum end of the century projected changes in PIC export (-0.40 PgC yr$^{-1}$) and the rain ratio (-0.053) under SSP5-8.5 are
smaller than the inter-model ranges of pre-industrial PIC export and rain ratios across the CMIP6 ensemble (respectively from
0.40 to 1.18 PgC yr$^{-1}$, and from 0.040 to 0.132).

The divergent PIC export projections are essentially explained by UKESM1-0-LL, GFDL-CM4 and GFDL-ESM4 (Fig. 1a).
These are the only ESMs that include a linear dependency of PIC production on the local saturation state (see the description
of their biogeochemical models, MEDUSA-2.1, BLINGv2 and COBALTv2, in Planchat et al., 2023). Through the absorption
of anthropogenic carbon, the upper ocean acidifies, the saturation state decreases, and calcification therefore declines in these
ESMs over SSP5-8.5 (respectively by -71.4 %, -73.5 % and -73.1 % in 2081-2100; Fig. 1a). Interestingly, for the only ESM in
CMIP6 that represents both implicit calcite and aragonite production, GFDL-ESM4, the decline in aragonite export (-80.3%)
is only slightly higher than that of calcite export (-71.6%), despite the higher solubility of aragonite.

For the other ESMs, PIC export trends are less divergent but even the sign of change is unclear, with ESMs projecting
increases of up to +22.8 % and decreases of up to -17.4 % in 2081-2100. For these ESMs, there is a positive correlation
between changes in PIC and POC export anomalies ($R^2$ = 0.65, p < 0.01, Fig. 3a). This is unsurprising given PIC production
is typically parameterized using a production ratio based on POC production (Planchat et al., 2023). As a result, the spatial
patterns of PIC and POC export changes are also highly positively correlated in 2081-2100, with general declines at low
latitudes partially offset by increase at high latitudes (see Fig. B2b,c,d and Fig. B3c,d,e).

## 3.2    Limited 21$^{st}$ century impact of PIC export on ocean carbon uptake and acidification

### 3.2.1    Insights from NEMO-PISCES sensitivity simulations

The NEMO-PISCES simulations were designed to assess the potential impact of CMIP6 carbonate pump biases and projection
uncertainties (see Table A2). The simulated PIC export is 1.16 PgC yr$^{-1}$ in the standard simulation and 0.81 PgC yr$^{-1}$ in the
carb_low simulation under pre-industrial conditions, in both cases remaining almost unchanged under increasing atmospheric
$CO_2$ concentration from 1850 to 2100. In contrast, and as expected from the simulation set-up (See Methods), PIC export
declines in carb– from 1.16 PgC yr$^{-1}$ to 0.87 PgC yr$^{-1}$ in 2081-2100 (-25 %), while in carb+ it increases to 1.50 PgC yr$^{-1}$
(+29 %). The difference between the standard and carb_low PIC export values encompasses CMIP6 biases in steady state
PIC export values, while the PIC export anomalies simulated in carb– and carb+ encompass CMIP6 anomalies under SSP5-
8.5 (Fig. 2a). In all NEMO-PISCES simulations, the POC export is identical and near constant over the period 1850 to 2100
(8.50 PgC yr$^{-1}$).





**Figure 1.** ESM projections of PIC export, ocean carbonate chemistry and anthropogenic carbon uptake. ESM projected anomalies relative to the pre-industrial control simulation in (a) PIC export at 100 m, (b) the rain ratio at 100 m, (c) surface sAlk, (d) anthropogenic carbon uptake, and (e) surface calcite saturation state in the Historical and SSP5-8.5 simulations. For each panel, the data are smoothed with 11-year rolling means, and the number of ESMs available, means, quartiles and extreme values in 2100 for both SSP5-8.5 and SSP1-2.6 are provided. Assessments of the 2021 anthropogenic ocean carbon uptake based on dataproducts and global ocean biogeochemical models (GOBMs), with their associated standard deviation is provided (GCP, Friedlingstein et al., 2022).



By maintaining pre-industrial ocean temperature and circulation throughout the NEMO-PISCES simulations, surface sAlk
responds directly to the changes in PIC export and not to other processes (Fig. 3b). Due to the effect of calcification on Alk,
sAlk increases when PIC export declines and vice versa. Thus, using carb– and carb+, a characteristic global relationship can
be inferred between PIC export and surface sAlk anomalies (slope: -5.12 x $10^{-2}$ mol m$^{-3}$ (PgC yr$^{-1}$)$^{-1}$, $p < 0.01$; Fig.3b).

The relative effect of changes in PIC export on 2081-2100 projections of anthropogenic ocean carbon uptake is small
(+0.25 PgC yr$^{-1}$ for carb– and -0.26 PgC yr$^{-1}$ for carb+ ; +4.0 % and -4.1 %, respectively) and near negligible with regard to
surface $\Omega_{calc}$ (+0.039 for carb– and -0.039 for carb+ ; +1.6 % and -1.6 %, respectively; Fig.3c,d). Yet, by constraining the PIC
export change in response to rising atmospheric $CO_2$ concentration in carb– and carb+, a high-end assessment of the carbonate
pump impact is obtained. Indeed, a decrease in PIC production would lead to a relative basification of the surface ocean, and
thus to a dampening of the effect related to the increase in atmospheric $CO_2$ concentration, i.e. a negative feedback. It should
be noted though that the relative change in anthropogenic carbon uptake is of the same order of magnitude as the absolute
change in PIC export (e.g. 0.25 PgC yr$^{-1}$ versus 0.29 PgC yr$^{-1}$ for carb–), illustrating a simple quantitative relationship between
PIC export changes and carbon uptake.

The impact of a negative bias in the mean-state PIC export (carb_low compared to standard) is limited for anthropogenic
ocean carbon uptake (+0.13 PgC yr$^{-1}$; +2.0 %), and relatively small with respect to surface acidification (+0.125 for $\Omega_{calc}$;
+5.2 %; Fig.3c,d). Indeed, at global scale, for the same atmospheric $CO_2$ concentration increase, the associated rise in DIC is
more important for carb_low than the standard simulation. This is essentially driven by a greater ratio between surface DIC
and the Revelle factor (Revelle and Suess, 1957) – directly connecting an absolute change in surface DIC to a relative change
in atmospheric $CO_2$ – for carb_low compared to the standard simulation (+2.7 %). As surface sAlk remains stable for these
two simulations, this leads to a higher carbon uptake, and an associated greater acidification over the period compared to the
standard simulation (Fig. 3b).

Climate change feedbacks on ocean carbon uptake and surface acidification far outweigh the acidification-driven change of
the carbonate pump (see Fig. B4e,f). When including all climate change effects, and in particular changes in ocean temper-
ature and circulation (standard_dyn+atm simulation), the anthropogenic carbon uptake decreases by 1.38 PgC yr$^{-1}$ and $\Omega_{calc}$
increases by 0.363 in 2081-2100 relative to the standard simulation (respectively -19.6 % and +15.2 %). This response has
been extensively studied using a variety of ocean-only models and ESMs for more than two decades (Sarmiento et al., 1998;
Schwinger et al., 2014; Arora et al., 2020; Canadell et al., 2021). These effects result from a combination of climate-change
impacts on anthropogenic carbon uptake through changes in the physical (main driver) and biological pumps with decreased
solubility and increased stratification / decreased ventilation (e.g. Canadell et al., 2021).

Building on the NEMO-PISCES sensitivity simulations, we can assess the potential extent to which PIC export anomalies
may influence surface sAlk, anthropogenic carbon uptake and surface acidification in CMIP6 ESM projections, and over what
time scales.



**Figure 2.** NEMO-PISCES sensitivity simulations. Absolute values of (a) PIC export at 100 m, (b) surface sAlk, (c) anthropogenic carbon uptake and (d) surface calcite saturation state. For each panel, the data are smoothed with 11-year rolling means, and CMIP6 ensemble statistical elements (mean, quartiles and extreme values) are provided for early Historical experiment values in 1850. The CMIP6 ensemble anomaly mean (black line) and range (grey shading) are shown bias corrected to the 1850 value of the NEMO-PISCES standard simulation.



### 3.2.2 Minimal influence in CMIP6 ESM projections

The analysis of the CMIP6 ensemble confirms the robust negative relationship between the PIC export and surface sAlk anomalies by the end of the century (Fig. 3b). As with projections of PIC export, UKESM1-0-LL, GFDL-CM4 and GFDL-ESM4 stand out with significant increases in surface sAlk (respectively +0.022 mol m$^{-3}$, +0.021 mol m$^{-3}$ and +0.021 mol m$^{-3}$) in 2081-2100 under SSP5-8.5 (Fig. 1c). For the other ESMs, the sign of the anomaly remains uncertain, showing either an increase (up to +0.007 mol m$^{-3}$) or a decrease (up to -0.009 mol m$^{-3}$). At the end of the century, a variation in the amplitude of the PIC export is matched by a proportional variation in surface sAlk (slope: -6.09 x 10$^{-2}$ mol m$^{-3}$ (PgC yr$^{-1}$)$^{-1}$, $R^2$ = 0.97, p < 0.01), similar to relationship identified across the NEMO-PISCES simulations (Fig. 3b). Even when excluding ESMs with a large decrease in PIC export, we still find a significant correlation between PIC export changes and surface sAlk changes ($R^2$ = 0.80, p < 0.01). Thus, the divergent PIC export projections drive contrasting trends in surface sAlk across the CMIP6 ensemble, with ESMs that project a large decrease in PIC export consistently projecting increases in surface sAlk. PIC export, in addition to being the main driver of the quasi-steady state vertical sAlk gradient (Planchat et al., 2023), therefore also drives surface sAlk anomaly projections this century. As a result, there is some confidence in using observations of salinity-normalized surface Alk to identify trends in the PIC export at 100 m., and thus of the carbonate pump up until at least 2100 (Ilyina et al., 2009).

Interestingly, this relationship is robust despite the effect of POC export and circulation changes on Alk (see Appendix B1). Although the POC export anomaly in ESMs is, in absolute terms, 0.7 to 35 times larger than the PIC export anomaly (see Fig. B3b compared to Fig. 1a), changes in PIC export still drive surface sAlk anomalies. As a global decrease in POC export is projected in the CMIP6 ensemble, the effect of this change in the soft-tissue pump on surface sAlk could explain the negative intercept of the relationship obtained for the ESMs, since a negative POC export anomaly would drive a negative surface Alk anomaly (see Appendix B1). In addition, the positive relationship between PIC and POC exports (Fig. 3a) would tend to slightly increase the negative slope of the sAlk - PIC export relationship, as a decline in POC export would act to enhance declines in PIC export resulting in a greater decrease in surface sAlk (see Appendix B1b). Thus, the representation of the soft-tissue pump, from organic matter production to remineralization, has limited effects on the relationship between PIC export and surface sAlk anomalies.

ESMs also show differences in carbon uptake and surface ocean acidification from 1850 to 2100 (Fig. 1d,e). The trend shown is consistent for the CMIP6 ensemble although a dispersion of the ESMs persists and increases over the century under SSP5-8.5, especially for the anthropogenic carbon flux. The values for the anthropogenic carbon uptake in 2081-2100 under SSP5-8.5 range from 5.29 PgC yr$^{-1}$ to 6.99 PgC yr$^{-1}$, and for $\Omega_{calc}$ they range from -2.87 to -2.46. These ranges are larger than that obtained for our idealized sensitivity simulations, ranging from 6.77 PgC yr$^{-1}$ to 7.29 PgC yr$^{-1}$ for the anthropogenic carbon uptake, and from 2.35 to 2.43 for $\Omega_{calc}$. While these signals are mostly affected by the ability of the surface ocean to respond to the increase in atmospheric $CO_2$, there is no inter-ESM correlation between PIC export anomalies and either ocean carbon uptake or surface ocean acidification. Thus, only a limited fraction of the differences in anthropogenic carbon uptake and surface ocean acidification in the CMIP6 ensemble can be explained by changes in the carbonate pump.



**Figure 3.** Relationships between PIC export and POC export, ocean carbon uptake, surface alkalinity and acidification. Simulated PIC export anomalies at 100 m and the corresponding (a) POC export anomalies at 100 m (b) surface sAlk anomalies, (c) ocean anthropogenic carbon uptake and (d) surface calcite saturation state. Anomalies are calculated in 2081-2100 relative to the piControl experiments. Linear regressions of NEMO-PISCES simulations (dashed grey lines) are calculated using the standard, carb– and carb+ simulations (excluding carb_low, with a black pointer outline). CMIP6 ensemble linear regressions (solid grey lines) are calculated excluding ESMs with a black pointer outline, due to their PIC production saturation state dependency (a), and their surface sAlk drift or their omission of the influence of the soft tissue pump on sAlk (b; see Sect. 2.3 and Appendix B1).





### 3.3 Enhanced post 2100 impact of PIC export anomalies on the ocean carbon cycle

255 The relationship between PIC export and surface sAlk anomalies in the extended NEMO-PISCES simulations substantially changes post 2100 (Fig. 4a). While PIC export declines have a slightly greater impact on changes in sAlk (slope: -6.84 x $10^{-2}$ mol m$^{-3}$ (PgC yr$^{-1}$)$^{-1}$ in 2270-2299 compared to -5.12 x $10^{-2}$ mol m$^{-3}$ (PgC yr$^{-1}$)$^{-1}$ in 2081-2100), what is the most striking is a general increase in surface sAlk. This increase in sAlk even occurs for the standard and carb_low simulations that exhibited near-stable surface sAlk up until 2100 (+0.056 mol m$^{-3}$ for standard and +0.038 mol m$^{-3}$ for carb_low from 2100 260 to 2300; +2.4 % and +1.6 %). As a result, the intercept of the regression between PIC export and surface sAlk anomalies is shifted towards higher values by 2300 (from 1.02 x $10^{-3}$ mol m$^{-3}$ in 2081-2100 to 5.27 x $10^{-2}$ mol m$^{-3}$ in 2270-2299).

This general increase in surface sAlk responds to an abrupt shift in the vertical CaCO$_3$ dissolution pattern following the shoaling of the saturation horizon in response to long-term ocean acidification (Fig. 4b,c). While dissolution is mainly confined to deep waters (> 1500 m) and essentially the sediment interface prior to 2100, by around 2150 it almost entirely occurs in the 265 subsurface (< 1500 m) under high-emissions. This dissolution predominately occurs in the water column and no longer at the seafloor. The dependence of dissolution on the saturation state in NEMO-PISCES (Aumont et al., 2015; Planchat et al., 2023), therefore drives a sudden shift in PIC dissolution depth, impacting surface sAlk. Indeed, the shift is the consequence of the strong anthropogenic ocean acidification signal that slowly propagates towards the bottom from the surface. The subsurface ocean thus tends to be undersaturated before part of the ocean interior. A general increase in surface Alk has previously been 270 reported in other model simulations, but was attributed to increased seafloor dissolution due to ocean acidification (Ilyina et al., 2009) rather than a shift from seafloor to water column dissolution.

This abrupt shift in the vertical CaCO$_3$ dissolution pattern occurs regardless of the PIC export magnitude at quasi-steady state and its projected anomaly. Indeed, for all the NEMO-PISCES sensitivity simulations, it occurs at the very beginning of the 21$^{st}$ century, when the global scale mean of $\Omega_{calc}$ at 500 m reaches 1.23±0.01. This calcite saturation state threshold is 275 independent on the atmospheric CO$_2$ growth rate, ocean circulation and the carbonate pump magnitude. It is reached between 2100 (carb_low) and 2109 (carb–), respectively corresponding to an atmospheric CO$_2$ of 936 ppm and 1021 ppm. A drastic reduction in PIC production does not itself significantly increase the atmospheric CO$_2$ threshold (1021 ppm for carb– versus 1002 ppm for standard), but the smaller the steady-state carbonate pump is, the lower the threshold is (936 ppm for carb_low). Finally, considering also the effects of climate change on ocean circulation (standard_dyn+atm simulation), the shift is slightly 280 delayed by a decade with an atmospheric CO$_2$ threshold of 1088 ppm, since its consideration dampens ocean acidification (see Fig. B4f). Although a calcite saturation state threshold can robustly be pointed out, the shift itself should be reversible regarding this environmental control parameter, and should remain dependent to it. If subsurface $\Omega_{calc}$ is back at higher values than the threshold, then the vertical PIC dissolution should shift back at depth, probably without hysteresis. Therefore, it seems difficult to speak of a bifurcation tipping point for the oceanic carbon cycle (Chen et al., 2021).

The reduction in deep ocean CaCO$_3$ dissolution results in deep ocean acidification, which could regionally occur prior to the penetration of substantial anthropogenic carbon into the ocean interior (Fig. 4b). Indeed, in addition to surface acidification (-2.43 in 2081-2100 and -3.60 in 2280-2299 for $\Omega_{calc}$), deep ocean acidification occurs post 2100 (-0.04 in 2081-2100 and





-0.29 in 2280-2299 for $\Omega_{calc}$ at 4750 m). In contrast to the direct acidification associated with ocean carbon uptake this indirect acidification rises from the deep ocean towards the surface (Fig. 4b). It could therefore cause earlier than expected deep ocean
acidification in regions such as the North Pacific, where the transport of anthropogenic carbon into the the deep takes hundreds to thousands of years (e.g. Levin and Le Bris, 2015).

Between 2100 and 2300, the simulated $CaCO_3$ cycle thus exerts a weak negative feedback on the surface ocean, slightly dampening acidification and increasing carbon uptake, but a positive feedback in the deep ocean, where it enhances acidification. However, it could also trigger a compensatory effect at the seafloor by deepening the lyscoline, and inducing a dissolution
of sedimentary $CaCO_3$ (not currently represented in NEMO-PISCES), dampening the acidification signal in the deep ocean. Moreover, the shoaling of $CaCO_3$ dissolution raises questions about the impact that this would have if there is a protective and/or ballast effect of $CaCO_3$ on organic matter (e.g. Klaas and Archer, 2002; Passow and De La Rocha, 2006; De La Rocha and Passow, 2007; Lee et al., 2009; Engel et al., 2009; Moriceau et al., 2009). If such an effect is confirmed, enhanced subsurface PIC dissolution would increase the remineralisation of subsurface POC, possibly decreasing carbon uptake and dampenig
acidification.

Post 2100 changes in the carbonate pump are heightened, and regional variability in the air-sea carbon flux is enhanced (Fig. 5). Although the sAlk and $\Omega_{calc}$ anomalies associated with changes in PIC export at 100 m are quite homogeneous, with enhanced anomalies at low to mid latitudes, the air-sea carbon flux response is local and strongly connected to PIC export anomalies. In 2081-2100, in a region where PIC production and thus PIC export decline, there is enhanced ocean carbon uptake.
However, in 2280-2299, alongside the increase in ocean carbon uptake due to a more pronounced decline in PIC export, there are also regions of reduced ocean carbon uptake, notably in the Southern Ocean. Up to 2100, changes in the surface carbonate pump are the dominant driver of changes in ocean carbon uptake, acidification and sAlk (Fig. 5a,b,c,d). However, by 2300, the impact of upwelled deep waters that have experienced less PIC dissolution is also evident in high latitudes (Fig. 5e,f,g,h). This results in high latitude declines in ocean carbon uptake which explains why, beyond 2150, the anthropogenic carbon uptake
difference between carb– and carb+ is reduced (see Fig. B4e).

## 4 Conclusion and perspectives

The projected 21[st] century response of $CaCO_3$ export in the CMIP6 ensemble is highly divergent under high emissions (ranging from -74 % to +23 %). The ESMs with the largest export declines (< -70 %) all parameterize pelagic $CaCO_3$ production as a function of saturation state, highlighting the need for parameterization consensus to constrain PIC export projections. $CaCO_3$
export also drives divergent projections of the rain ratio and salinity-normalized surface alkalinity across the CMIP6 ensemble. As there is a robust negative correlation between $CaCO_3$ export anomalies and salinity-normalized surface alkalinity in the CMIP6 ensemble, salinity-normalized surface alkalinity observations could be used to identify historical trends in PIC export.

Sensitivity simulations performed with the marine biogeochemical model NEMO-PISCES confirm the limited impact of the carbonate pump on anthropogenic carbon uptake and ocean acidification in the 21[st] century. Improving the realism of the
representation of this carbon pump therefore does not appear to be a priority for constraining uncertainties in anthropogenic







**Figure 4.** Multi-centennial effects of acidification on the global carbonate pump under high emissions for the NEMO-PISCES sensitivity simulations. (a) The relationship between PIC export anomalies at 100 m and surface sAlk anomalies in 2280-2299 and 2081-2100 (as in Fig. 3b). Linear regressions (dashed and dotted grey lines) are calculated using the standard, carb– and carb+ simulations (excluding carb_low, with a black pointer outline). (b) Global PIC dissolution with depth both within the water column (top) and at the seafloor (bottom) for the NEMO-PISCES standard simulation. Calcite saturation state contours are displayed at 0.8, 1.0 and 1.2. The depth of maximum total PIC dissolution (i.e. the sum of water column and benthic dissolution) below 100 m – as a high quantity of PIC is produced and dissolved in shallow waters in NEMO-PISCES – is shown in red. (c) Evolution of the total PIC dissolution profile in a high-emissions scenario with a mirrored bathymetric profile to highlight the extent of the seafloor according to depth.



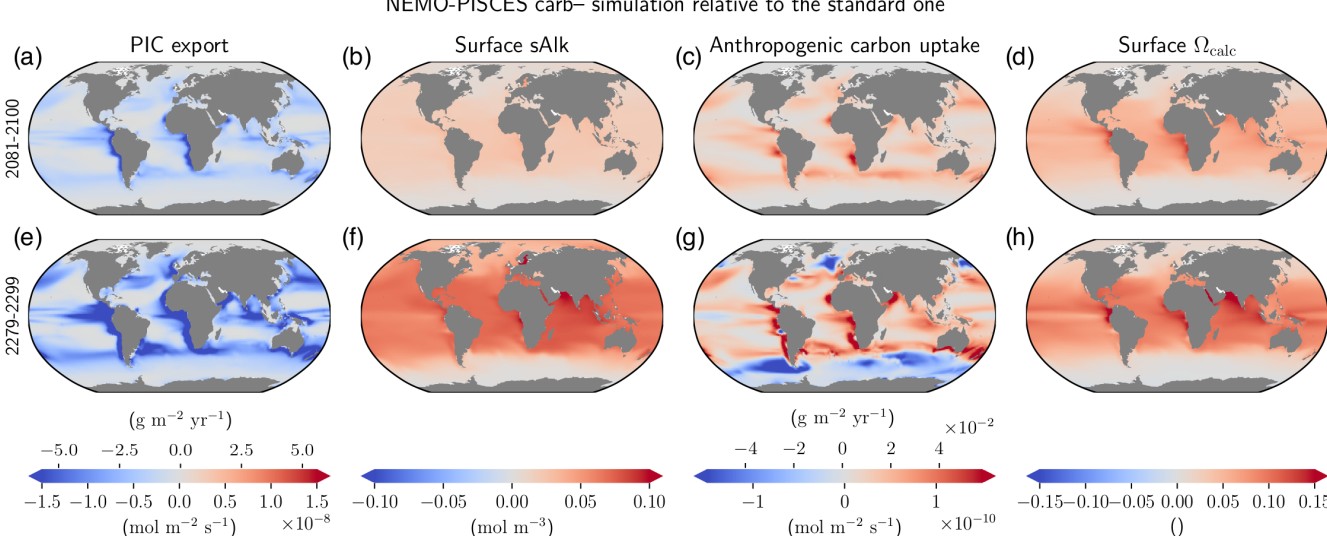

**Figure 5.** Regional differences in the multi-centennial impact of a declining carbonate pump on ocean carbon uptake and acidification. carb– anomalies of (a,e) PIC export at 100 m, (b,f) surface sAlk, (c,g) anthropogenic carbon uptake, and (d,h) surface $\Omega_{calc}$ relative to the standard simulation, in 2081-2100 (a,b,c,d) and 2280-2299 (e,f,g,h).

carbon uptake and ocean acidification. Nevertheless, on multi-centennial timescales, the effects of a perturbation in the carbonate pump on the carbon cycle are heightened. In particular, a global-scale threshold for the oceanic $CaCO_3$ cycle is highlighted with an abrupt change in the vertical pattern of $CaCO_3$ dissolution from the deep ocean to the subsurface when the global scale mean calcite saturation state reaches about 1.23 at 500 m (likely when atmospheric $CO_2$ reaches 900 to 1100 ppm). The

(sub)surface acidification signal can lead to a deep ocean $CaCO_3$-induced acidification signal post 2100 due to the collapse of $CaCO_3$ dissolution at depth. The effects of these changes on carbon uptake and surface ocean acidification remain limited, despite the appearance of regional variability in air-sea carbon fluxes, with upwelled deep waters oppositely impacted by a change in PIC export relative to surface waters.

At present, the physical carbon pump plays the main role in anthropogenic ocean carbon uptake and acidification (Canadell

et al., 2021). However, under high-emissions the physical carbon pump is likely to saturate before the end of the century due to declining ocean buffer capacity and reduced ocean ventilation (Canadell et al., 2021; Chikamoto et al., 2023). Changes in the biological carbon pump, including the carbonate pump, post 2100 are therefore likely to have more important feedbacks on anthropogenic carbon uptake and acidification. Although projected declines in the carbonate pump have limited impact on global climate and ocean acidification this century, such declines may still have important local ecosystem impacts.



# Appendix A: Methodology

## A1 CMIP6 ESMs

We share here details regarding the ESMs considered in this analysis (Table A1).

**Table A1.** Summary of the CMIP6 ESMs, their coupled marine biogeochemical models and the experiments considered in this intercomparison.

| Group | ESM | MBG | Variant label | Experiments |
|---|---|---|---|---|
| CCCma | CanESM5 | CMOC | r1i1p2f1 | piControl, Historical, SSP1-2.6, SSP5-8.5 |
| | CanESM5-CanOE | CanOE | r1i1p2f1 | piControl, Historical, SSP1-2.6, SSP5-8.5 |
| CMCC | CMCC-ESM2 | BFM5.2 | r1i1p1f1 | piControl, Historical, SSP1-2.6, SSP5-8.5 |
| CNRM-CERFACS | CNRM-ESM2-1 | PISCESv2-gas | r1i1p1f2 | piControl, Historical, SSP1-2.6, SSP5-8.5 |
| CSIRO | ACCESS-ESM1-5 | WOMBAT | r1i1p1f1 | piControl, Historical, SSP1-2.6, SSP5-8.5 |
| IPSL | IPSL-CM6A-LR | PISCESv2 | r1i1p1f1 | piControl, Historical, SSP1-2.6, SSP5-8.5 |
| MIROC | MIROC-ES2L | OECO2 | r1i1p1f2 | piControl, Historical, SSP1-2.6, SSP5-8.5 |
| MOHC | UKESM1-0-LL | MEDUSA-2.1 | r1i1p1f2 | piControl, Historical, SSP1-2.6, SSP5-8.5 |
| MPI-M | MPI-ESM1-2-LR | HAMOCC6 | r1i1p1f1 | piControl, Historical, SSP1-2.6, SSP5-8.5 |
| | MPI-ESM1-2-HR | HAMOCC6 | r1i1p1f1 | piControl, Historical, SSP1-2.6, SSP5-8.5 |
| MRI | MRI-ESM2-0 | NPZD-MRI | r1i2p1f1 | piControl, Historical, SSP5-8.5 |
| NCAR | CESM2-WACCM | MARBL | r1i1p1f1 | piControl, Historical, SSP1-2.6, SSP5-8.5 |
| NCC | NorESM2-LM | iHAMOCC | r1i1p1f1 | piControl, Historical, SSP1-2.6, SSP5-8.5 |
| NOAA-GFDL | GFDL-CM4 | BLINGv2 | r1i1p1f1 | piControl, Historical, SSP5-8.5 |
| | GFDL-ESM4 | COBALTv2 | r1i1p1f1 | piControl, Historical, SSP1-2.6, SSP5-8.5 |

## A2 Conservation of the global Alk inventory

Historically, NEMO-PISCES was configured with annual Alk restoration to keep its global inventory constant. This was notably the case with PISCESv1 for CMIP5 (IPSL-CLM5A-LR/MR, and IPSL-CM5B-LR) and with PISCESv2 for CMIP6 (IPSL-CM6A-LR). Although the balance of the global Alk inventory may be questioned at pre-industrial state (Planchat et al., 2023), the use of an Alk restoration term may hide an imbalance in biogeochemical processes and may impact the carbon cycle. We therefore constrained the conservation of the global Alk inventory keeping a degree of freedom for the carbonate pump at the seafloor. Indeed, during the spin-up, at each time step, a coefficient is calculated at the global scale to adjust the fraction of PIC reaching the seafloor that is buried to counterbalance the net sink of Alk resulting from all the other biogeochemical processes affecting it (Fig. A1). After 2500 yr of spin-up, we estimated the drift of this seafloor dissolution parameter low enough to fix it at the value obtained at the end of the spin-up. The strategy employed here – broadly similar to that developed



in MARBL by NCAR (Long et al., 2021) and in COBALTv2 by NOAA-GFDL (Dunne et al., 2012) – only allows the global
Alk inventory to be conserved relative to the biogeochemical processes that affect it. Surface dilution/concentration, as well as

few physical processes, which are not perfectly conservative in the model, lead to a slight negative Alk drift in the configuration
used (< 0.03 PgC yr$^{-1}$ in absolute terms), which is nevertheless of the same order of magnitude as the global net sink of the
soft-tissue pump on Alk.

## A3   RCP versus SSP

The use of the RCP-8.5 scenario for the NEMO-PISCES sensitivity analysis is considered as a proof of concept, being slightly

less extreme than the SSP5-8.5 scenario (Fig. A2).

## A4   NEMO-PISCES sensitivity simulations

We share here details regarding the sensitivity simulations carried out with NEMO-PISCES in this analysis (Table A2).

**Table A2.** Summary of the sensitivity simulations led with NEMO-PISCES. The key elements of these simulations, especially the forcing
and parameters used are shared for each of the experiments.

| Simulation | Dates (duration) | Atmospheric $CO_2$ | Ocean dynamics | Production ratio parameter | Fraction dissolved at the seafloor |
|---|---|---|---|---|---|
| standard_spinup | 1800-2299 (2500 yr) | Pre-industrial | Pre-industrial | 0.45 | Free |
| standard_picontrol | 1800-2299 (500 yr) | Pre-industrial | Pre-industrial | 0.45 | 0.934 |
| standard | 1852-2299 (448 yr) | Historical+RCP-8.5 | Pre-industrial | 0.45 | 0.934 |
| carb– | 1852-2299 (448 yr) | Historical+RCP-8.5 | Pre-industrial | 0.45; decreased with atmospheric $CO_2$ | 0.934 |
| carb+ | 1852-2299 (448 yr) | Historical+RCP-8.5 | Pre-industrial | 0.45; increased with atmospheric $CO_2$ | 0.934 |
| standard_dyn | 1852-2299 (448 yr) | Pre-industrial | Historical+RCP-8.5 | 0.45 | 0.934 |
| standard_dyn+atm | 1852-2299 (448 yr) | Historical+RCP-8.5 | Historical+RCP-8.5 | 0.45 | 0.934 |
| carb_low_spinup | 1800-2299 (2500 yr) | Pre-industrial | Pre-industrial | 0.30 | Free |
| carb_low_picontrol | 1800-2299 (500 yr) | Pre-industrial | Pre-industrial | 0.30 | 0.789 |
| carb_low | 1852-2299 (448 yr) | Historical+RCP-8.5 | Pre-industrial | 0.30 | 0.789 |

## A5   Surface sAlk drifts

The surface sAlk faces a non-negligible drift for two ESMs in the CMIP6 ensemble, CMCC-ESM2 and CNRM-ESM2-1

(Fig. A3a). While this drift is directly associated with a surface Alk drift for CNRM-ESM2-1 (Fig. A3b), it is linked to a
salinity drift in the case of CMCC-ESM2 (Fig. A3d), also impacting sDIC as a consequence (Fig. A3c).



**Figure A1.** Alk sinks and sources in NEMO-PISCES pre-industrial standard simulation. (a) Integrated Alk sinks (negative) and sources (positive) associated with all the biogeochemical processes affecting it, which can occur in the water column, at the seafloor or the surface, or along the coasts. The colors refer to the type of processes affecting Alk associated with: (i) the carbonate pump through calcite production (red) and dissolution (orange); (ii) the soft-tissue pump through organic matter production (blue), organic matter remineralization (purple) and nitrogen reactions (cyan); and (iii) the boundary fluxes through riverine inputs and dust deposition (green). The net Alk budget for these three different components is shared in the bottom sub-panel. (b) Vertical distribution of the global-scale area-weighted average of the various Alk sinks/sources, with a zoom on the first 500 m in the top sub-panel.





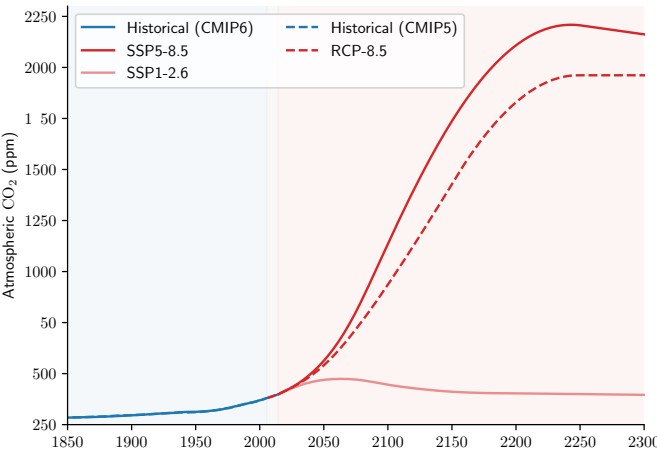

**Figure A2.** RCP versus SSP. Evolution of the atmospheric $CO_2$ concentration according to the scenarios considered in our analysis: (i) Historical (CMIP6), SSP5-8.5, and SSP1-2.6 for the CMIP6 ensemble, and (ii) Historical (CMIP5), and RCP-8.5 for the NEMO-PISCES sensitivity simulations.

**Appendix B: Results and discussion**

**B1   Surface sAlk anomaly in 2081-2100**

We share here a theoretical decomposition of the surface sAlk anomaly in 2081-2100 for the CMIP6 ensemble from the PIC
and POC export anomalies at 100 m. To perform this decomposition, we use the relationship between the PIC export anomaly
at 100 m and the surface sAlk anomaly that we obtained with the NEMO-PISCES sensitivity simulations. We use the slope of
this linear regression ($-5.12 \times 10^{-2}$ mol m$^{-3}$ (PgC yr$^{-1}$)$^{-1}$) to construct an indicative first-order transformation of the impact of
a carbon particle export anomaly at 100 m on surface sDIC; a sort of conversion from petagrams of carbon per year to moles
per cubic meter. We will note this parameter as $c_0$. Since Alk increases for a decrease in PIC export and is twice as impacted
by a change in PIC export compared to DIC, $c_0 = -1 \times 0.5 \times (-5.12 \times 10^{-2}) = 2.06 \times 10^{-2}$ mol m$^{-3}$ (PgC yr$^{-1}$)$^{-1}$. Then, the surface
sAlk anomaly resulting from a PIC export anomaly at 100 m can be expressed as:

$$\Delta \text{sAlk}_{\text{PIC}} = -\frac{2}{1} \cdot c_0 \cdot \Delta \text{Ex(PIC)}^{100 \text{ m}}. \tag{B1}$$

Similarly, the surface sAlk anomaly relative to a POC export anomaly at 100 m can be expressed as:

$$\Delta \text{sAlk}_{\text{POC}} = -\frac{-r_{\text{N:P}}}{r_{\text{C:P}}} \cdot c_0 \cdot \Delta \text{Ex(POC)}^{100 \text{ m}}, \tag{B2}$$

where we consider that POC export anomalies induce surface anomalies with an ESM-dependent ratio equal to $-r_{\text{N:P}}$:$r_{\text{C:P}}$
(the opposite of the nitrogen to phosphorus ratio divided by the carbon to phosphorus ratio; see Planchat et al., 2023, their
Supplement Table S1 for the associated values per ESM). In particular, the sAlk anomaly associated with POC is null for
UKESM1-0-LL, since the soft tissue pump does not affect Alk in MEDUSA-2.1 (see Planchat et al., 2023, their Supplement





**Figure A3.** Surface sAlk drift, causes and consequences. Time series of the piControl experiment (a) surface sAlk, (b) surface sDIC, (c) surface Alk and (d) surface salinity for all the CMIP6 ensemble, as well as the ensemble mean.

Table S1). Thus, as a first approximation, we can write a theoretical decomposition of the surface sAlk anomaly as:

$$\Delta sAlk \approx \Delta sAlk_{PIC} + \Delta sAlk_{POC} \tag{B3}$$


This decomposition is idealised. First, we consider that changes in the export at 100 m reflect surface sAlk changes. Second, the way we consider the effect of the POC export anomaly on sAlk is simplified, not taking into account the potential decoupling between the remineralisation and the nitrogen reactions. Third, it does not take into account the consequences of the envisaged slowing of the meridional overturning circulation (MOC), which could have an impact on the surface sAlk (e.g. through an

attenuation of the upwelling of Alk-enriched deep waters, Chikamoto et al., 2023). Nevertheless, it highlights the primary role of the PIC export anomaly in driving the surface sAlk anomaly by 2100 (Fig. B1a), the POC export anomaly influence being second order (Fig. B1b). The combination of these two anomaly sources gives a result that is consistent with the modelled





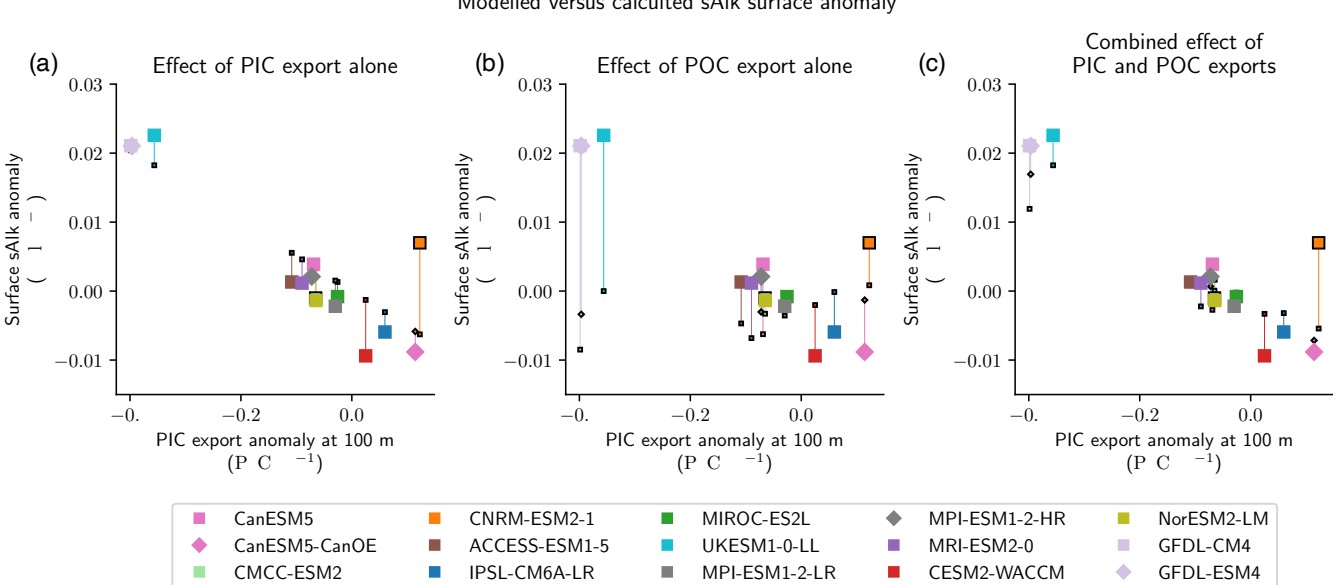

**Figure B1.** Theoretical decomposition of the sAlk surface anomaly. Respective effect of the PIC (a) and POC (b) export anomalies at 100 m on the surface sAlk anomaly in 2081-2100. The combined effect of both export anomalies is displayed in (c). For each of the three panels, in addition to the modelled values (big colored squares and diamonds), the associated calculated values are displayed in small with black pointer outlines and linked by colored lines.

anomaly values in the CMIP6 ensemble in 2081-2100 (Fig. B1c). The fit can be further improved by increasing $c_0$, which is realistic, as climate change induces reduced ventilation, and thus a relative stagnation of surface waters, contrary to the
idealised offline simulations we carried out with NEMO-PISCES, where only the concentration of atmospheric $CO_2$ varied. Two models can be singled out though. While CNRM-ESM2-1 stands out from the others due to its surface Alk drift (see Fig. A3a), CESM2-WACCM certainly stands out due to its slow circulation bias (Frischknecht et al., 2022), which would tend to accentuate the impact of a carbon export anomaly on the surface sAlk.

## B2 Temporal and spatial trends

We share here additional information regarding PIC and POC export trends in the CMIP6 ensemble (Fig. B2 and B3).

## B3 Sensitivity simulations post 2100

We share here the extended time series and the full set of sensitivity simulations completed with NEMO-PISCES for the key variables considered in this analysis.

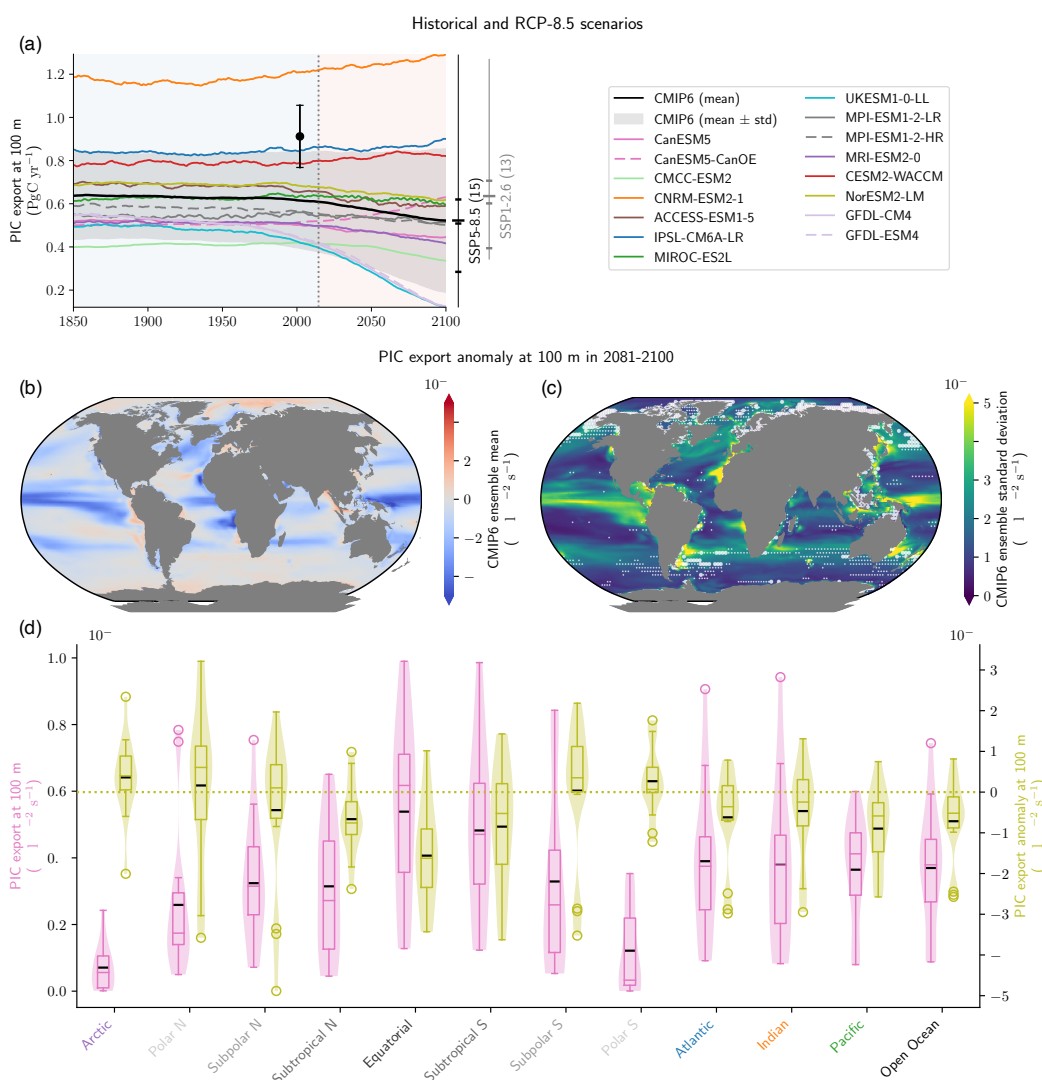

**Figure B2.** PIC export trends in the CMIP6 ensemble (supplement to Fig. 1a). (a) Absolute values of PIC export at 100 m for the Historical and SSP5-8.5 experiments. The black dot refers to the observationally based estimate from Sulpis et al. (2021) – although referenced to 300 m – with its associated uncertainty. In (a), the data are smoothed with 11-year rolling means, and the number of ESMs available, means, quartiles and extreme values in 2100 for both SSP5-8.5 and SSP1-2.6 are provided. (b,c) PIC export anomaly at 100 m in 2081-2100 relative to pre-industrial state for (b) the CMIP6 ensemble mean and (c) the CMIP6 standard deviation. In (c), partly transparent white circles points towards regions where strictly less than 5 ESMs (big circles) or strictly less than 7 ESMs (small circles) agree on the sign of change with the CMIP6 ensemble mean. This white shading highlights areas where few ESMs might strongly affect the CMIP6 ensemble mean value. (d) Violin plot of the PIC export at 100 m distribtion at pre-industrial state (pink) and for its anomaly in 2081-2100 (green) for specific open ocean basins (see Planchat et al., 2023, their Fig. A1).



**Figure B3.** POC export trends in the CMIP6 ensemble (as for PIC export in Fig. B2 and Fig. 1a). (a) Absolute values of POC export at 100 m for the Historical and SSP5-8.5 experiments. The black dot refers to the observationally based estimate from DeVries and Weber (2017). (b) ESM projected anomalies relative to pre-industrial control simulation in POC export at 100 m in the Historical and SSP5-8.5 simulations. In (a,b), the data are smoothed with 11-year rolling means, and the number of ESMs available, means, quartiles and extreme values in 2100 for both SSP5-8.5 and SSP1-2.6 are provided. (c,d) POC export anomaly at 100 m in 2081-2100 relative to pre-industrial state for (c) the CMIP6 ensemble mean and (d) the CMIP6 standard deviation. In (d), partly transparent white circles points towards regions where strictly less than 5 ESMs (big circles) or strictly less than 7 ESMs (small circles) agree on the sign of change with the CMIP6 ensemble mean. This white shading highlights areas where few ESMs might strongly affect the CMIP6 ensemble mean value. (e) Violin plot of the POC export at 100 m distribtion at pre-industrial state (pink) and for its anomaly in 2081-2100 (green) for specific open ocean basins (see Planchat et al., 2023, their Fig. A1).

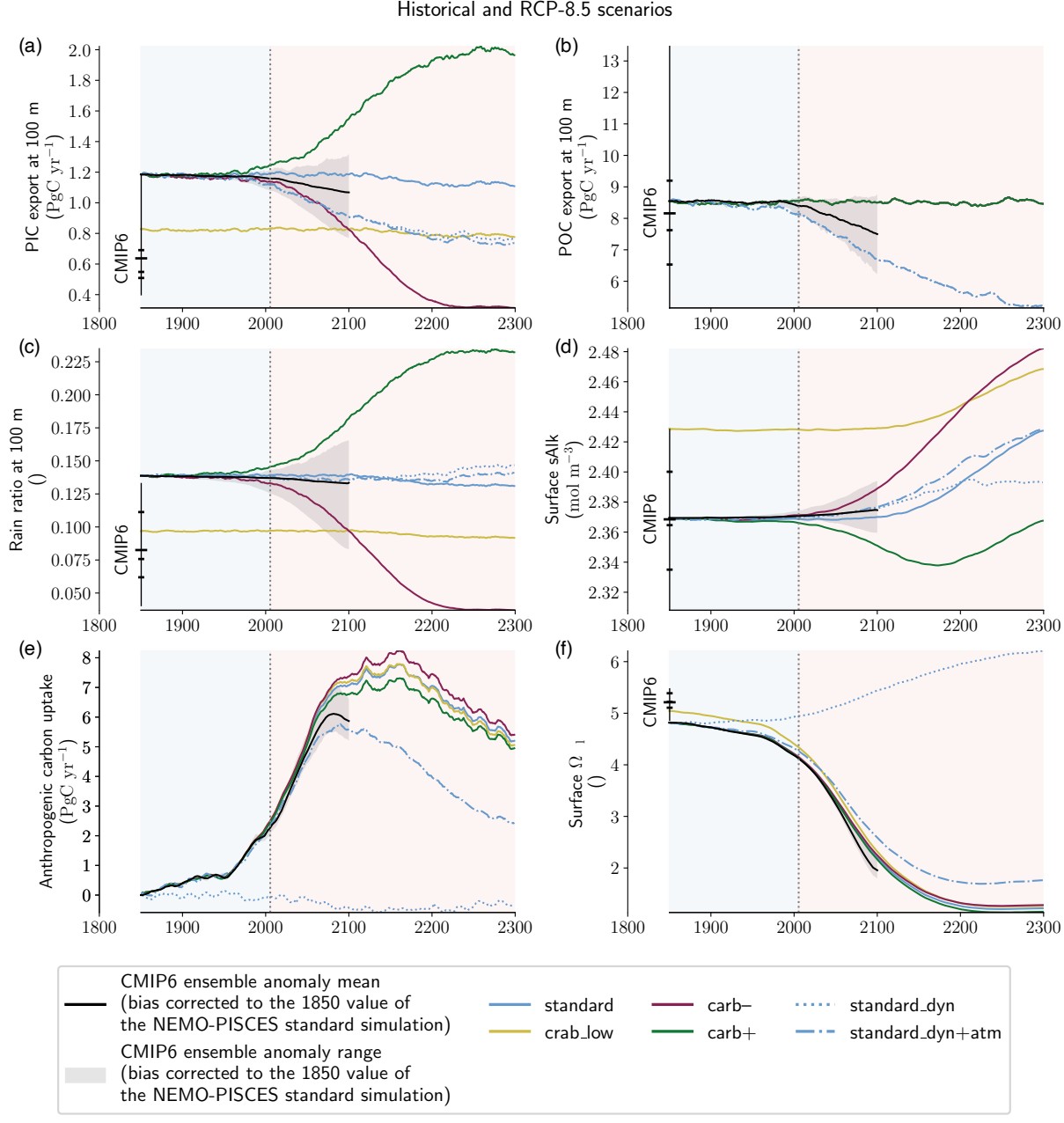

**Figure B4.** Sensitivity simulations with NEMO-PISCES (full time series and full set of NEMO-PISCES simulations; see Table A2; supplement to Fig. 2). Absolute values of (a) PIC export at 100 m, (b) POC export at 100 m, (c) the rain ratio at 100 m, (d) surface sAlk, (e) anthropogenic carbon uptake, and (f) surface calcite saturation state for all the sensitivity simulations. For each panel, the data are smoothed with 11-year rolling means, and CMIP6 ensemble statistical elements (mean, quartiles and extreme values) are provided for early Historical experiment values in 1850. The CMIP6 ensemble anomaly mean (black line) and range (grey shading) are shown bias corrected to the 1850 value of the NEMO-PISCES standard simulation.

*Code availability.* NEMO is released under the terms of the CeCILL licence. The standard NEMO-PISCES version (PISCESv2; Aumont et
al., 2015) slightly modified in this study (see Sect. 2.2) is accessible through http://forge.ipsl.jussieu.fr/igcmg_doc/wiki/Doc/Config/NEMO.
The other NEMO-PISCES versions are available on request from the corresponding author.

*Data availability.* All the CMIP ensemble data were available on at least one of the Earth System Grid Federation (ESGF) nodes.

*Author contributions.* This work is in the framework of the OMIP-BGC group, which contributed collectively to this study through the
organization and execution of the CMIP exercises and the sharing of simulation outputs. AP: conceptualization, investigation, methodology,
formal analysis, visualization, writing – original draft preparation – and project adminis- tration. LB and LK: supervision, funding acquisition,
methodology, resources, conceptualization, and writing – original draft preparation. OT: software.

*Competing interests.* The contact author has declared that none of the authors has any competing interests.

*Disclaimer.* This article reflects only the authors' views; the funding agencies and their executive agencies are not responsible for any use
that may be made of the information that the article contains.

*Acknowledgements.* We are grateful to the World Climate Research Programme's Working Group on Coupled Modelling, which is respon-
sible for the CMIP exercises. For CMIP, the U.S. Department of Energy's Program for Climate Model Diagnosis and Intercomparison
provided coordinating support and led the development of software infrastructure in partnership with the Global Organization for Earth
System Science Portals. This study benefitted from the ESPRI (Ensemble de Services Pour la Recherche à l'IPSL) computing and data
centre (https://mesocentre.ipsl.fr, last access: September 2022), which is supported by CNRS, Sorbonne Université, École Polytechnique,
and CNES and through national and international grants. We are also grateful to the administrative and technical staff at Ecole Normale
Supérieure/PSL.

*Financial supports.* Alban Planchat, Laurent Bopp, Lester Kwiatkowski, and Olivier Torres are grateful to the ENS-Chanel
research chair.



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
