# Peer review of "Carbonate pump feedbacks on alkalinity and the carbon cycle in the $21^{st}$ century and beyond"

_EGUsphere, 2023_

## Author Comment (AC1)

**Report on our manuscript revision**

*Dear Editor,*

*Thank you for your positive feedback on our manuscript.*

*We have carefully addressed the reviewers' comments in our revision and would like to express our gratitude to both reviewers for their meticulous review, which has significantly contributed to the enhancement of the manuscript.*

*Below, please find a detailed point-by-point response to all reviewers' comments.*

*Kind regards,*

*Alban Planchat, on behalf of all co-authors.*

**Responses to Referee #1 (anonymous)**

→ We thank Referee #1 for his/her comments and suggestions for improving our manuscript.

Review of "Carbonate pump feedbacks on alkalinity and the carbon cycle in the 21st century and beyond" by Planchat et al.

Using CMIP6 models, the authors explored how the biogenic CaCO3 export at a 100 m depth responds to projected increases in atmospheric CO2 and how the different responses impact ocean surface alkalinity, the saturation state with respect to calcite, and oceanic CO2 uptake during the 21st century. Motivated by the CMIP6 model spread in the projected CaCO3 export, the authors further explored the oceanic responses to imposed CaCO3 export changes for the extended time-period of 2100-2300 using an offline ocean biogeochemistry model. The authors nicely showed that the carbonate pump is one of the least constrained processes for the oceanic carbon pump and its projected uncertainty is very large. By assessing how the projected uncertainty can propagate into uncertainties in simulated oceanic carbon cycle on decadal to multi-centennial timescales, this study elucidates potential feedbacks from the carbonate pump on future carbon cycles. It is especially interesting to see that CaCO3 dissolution could respond abruptly to ocean acidification and that the sudden shift in CaCO3 dissolution could impact the regional patterns of oceanic CO2 uptake within the next century. The relatively minor effects of changing carbonate pumps on the oceanic CO2 uptake, compared to changing ocean physical dynamics, seem also novel and insightful. Overall, I have minor points.

1. The long-term response of CaCO3 dissolution to ocean acidification and its feedback on the Southern Ocean CO2 uptake seem very interesting. A question is how realistic this could be. Although a detailed model description is already presented in a previous publication by the same author, it might be worth highlighting here how the CaCO3 dissolution is parameterized in the PISCES model and how this parameterization compares with some observational constraints and/or models (e.g., Subhas et al. (2017); Liang et al. (2023) ). Related with this, the authors might say something about how CaCO3 dissolution is parameterized among the CMIP models in Introduction, as the effects of CaCO3 dissolution would become increasingly more important for the oceanic carbon cycle on longer timescales than the effects of CaCO3 export.

→ We now provide information for NEMO-PISCES concerning the consideration of saturation state dependency for the production and dissolution of CaCO3 in the methods:
"In NEMO-PISCES, PIC production does not depend on the local saturation state; however, PIC dissolution linearly depends on the saturation state, whether in the water column or at the ocean floor (Planchat et al., 2023)."
→ However, we do not wish to revisit in this paper the complexity of the carbonate pump parameterizations in ESMs, as already discussed in Planchat et al., 2023. We do however provide an overview of this in the introduction:
"Despite this, all current ESMs implicitly model CaCO3 production based on POC production, and rarely with a saturation-state dependency (Planchat et al., 2023). Models also typically consider calcite and not aragonite production, which may induce delays in the response of the carbonate pump to acidification, as aragonite is less stable than calcite. Similarly, models may underestimate carbonate pump feedbacks by not representing benthic calcifiers, such as corals, which are likely to be particularly vulnerable to climate change (Bindoff et al., 2019)."
We also specify the current status of considering a saturation state dependency for the production of CaCO3 in Sect. 3.1:
"The divergent PIC export projections are essentially explained by UKESM1-0-LL, GFDL-CM4 and GFDL-ESM4 (Fig. 1a). These are the only ESMs that include a linear dependency of PIC production on the local saturation state (see the description of their biogeochemical models, MEDUSA-2.1, BLINGv2 and COBALTv2, in Planchat et al., 2023)."

And we do the same regarding the dissolution of CaCO3 in Sect. 3.3:

"The dependence of dissolution on the saturation state in NEMO-PISCES (Aumont et al., 2015) – considered in about half of the CMIP6 ESMs (Planchat et al., 2023) – therefore drives a sudden shift in PIC dissolution depth, impacting surface sAlk."

→ A sentence has also been added in the section on post-2100 effects (Sect. 3.3) to underscore that the observed dissolution shift with NEMO-PISCES could be further accentuated if the parameterization of CaCO3 dissolution aligned with the results of laboratory studies, exhibiting a non-linear dependency (exponent > 1) on the saturation state in the water column:

"Furthermore, this shift could be even more abrupt, and CaCO3 dissolution further confined to surface waters if the saturation state dependency of CaCO3 dissolution were not linear, as suggested by laboratory studies, which indicate an exponent > 1 for PIC dissolution in the water column (e.g. Subhas et al., 2015)."

2. Please revise figures. In many figures (e.g., Figure 3), X- and Y-axes labels and units seem incomplete.

→ Indeed, this issue is noticeable in the file shared on ESD; however, the formatting issues do not appear in our PDF version. We will ensure that the problem is resolved in the final version shared on ESD.

Line #288-289: From Fig. 4b, I don't see that the indirect acidification rises from the deep to the surface. Instead, I see that the depth of CaCO3 dissolution (therefore basification) rises from the deep ocean towards the surface.

→ This indirect acidification signal at depth is apparent following the rise in the PIC dissolution negative anomaly at depth. This becomes particularly evident from 2100 when dissolution ascends rapidly in the subsurface, reducing dissolution at depth and thereby generating an acidification signal that ascends from the bottom to the surface (Fig. R1).

[Figure]

Figure R1: Supplement to Fig. 4b. Anomaly relative to 1850-1900 of the global PIC dissolution with depth both within the water column (left) and at the seafloor (right) for the NEMO-PISCES standard simulation. Calcite saturation state contours are displayed at 0.8, 1.0 and 1.2. The depth of maximum total PIC dissolution (i.e. the sum of water column and benthic dissolution) below 100 m – as a high quantity of PIC is produced and dissolved in shallow waters in NEMO-PISCES – is shown in red.

Line #299-300: Increasing subsurface POC remineralization itself would cause subsurface ocean acidification, opposing the effects of reduced CO2 uptake.

→ This has been rewritten:

*"If such an effect is confirmed, enhanced subsurface PIC dissolution would increase the remineralisation of subsurface POC, leading to subsurface ocean acidification, which would counteract the associated effect of reduced CO2 uptake."*

Data availability: Perhaps, the authors can make the NEMO-PISCES sensitivity experiments available to the public?

→ A Zenodo link is now provided in the 'Code availability' section to access the code associated with the sensitivity simulations conducted in this analysis.

References:

- Liang, H., Lunstrum, A. M., Dong, S., Berelson, W. M., & John, S. G. (2023). Constraining $CaCO_3$ export and dissolution with an ocean alkalinity inverse model. *Global Bigeochem. Cycles, 37*, e2022GB007535. https://doi.org/10.1029/2022GB007535
- Subhas, A. V., Adkins, J. F., Rollins, N. E., Naviaux, J., Erez, J., & Berelson, W. M. (2017). Catalysis and chemical mechanisms of calcite dissolution in seawater. *Proceedings of the National Academy of Sciences, 114*(31), 8175-8180. https://doi.org/doi:10.1073/pnas.1703604114

**Responses to Referee #2 (John Dunne)**

→ We thank Referee #1 for his/her comments and suggestions for improving our manuscript.

The manuscript "Carbonate pump feedbacks on alkalinity and the carbon cycle in the 21st century and beyond" by Planchat et al analyzes ocean carbonate cycling in CMIP6 ESMs and performs an additional idealized analysis with NEMO-PISCES simulations to explore the underlying mechanisms. The manuscript is an important addition quantifying trends and uncertainties from the CMIP6 generation ESMs on ocean carbonate cycling changes out to 2100 where the impact on carbon uptake is shown to be small, as well as post 2100 in which internal ocean changes to carbonate saturation states lead to fundamental shifts in carbonate cycling and impacts on surface alkalinity. I recommend publication with minor technical revision as outlined below.

Technical points:

20 – I think "basis" should be "base"
→ This has been corrected.

35 – "towards" should be "to" and ", which explains why it" should be "in what"
→ This has been corrected.

43-44 – I am not sure if this statement is appropriate as a blanket statement or only in the context of the steady state/preindustrial ocean. I also do not think the current Zeebe reference about Boron is relevant to the argument… perhaps the authors intended this one instead? Zeebe, R. E., & Wolf-Gladrow, D. (2001). CO2 in seawater: equilibrium, kinetics, isotopes (No. 65). Gulf Professional Publishing.
→ We have corrected the reference to Zeebe and Wolf-Gladrow (2001) throughout the entire paragraph and provided additional clarification on the application of the statement:
"For instance, relative anomalies of DIC associated with changes in pCO2 can be quantified using the Revelle factor, which can be expressed as a function of the rain ratio (Zeebe and Wolf-Gladrow, 2001), particularly for making large-scale order-of-magnitude assessments."

132 and Equation 1 – The text says that alpha is multiplied by the PIC production, but inspection of the equation for alpha looks like it is zero at preindustrial CO2 of 285 ppm which would mean zero PIC production… should "multiplied by" be "added to" – in which case alpha has units of PIC production, or perhaps the equation goes to 1 at preindustrial? Also, the authors should specify how they derived the value of 0.15.
→ This, as previously written, was inaccurate. To implicitly represent the production of CaCO3 derived from organic particle production in NEMO-PISCES, a production ratio parameter is considered (Planchat et al., 2023). In our standard simulation, this parameter is fixed at 0.45. However, in our carb+ and carb- simulations, it varies over time and is expressed as $0.45 \pm \alpha_{carb}$, where $\alpha_{carb}=0.15 \cdot (CO2-285)/(936-285)$. This adjustment allows for a 1/3 reduction in CaCO3 production by 2100, resulting in a CaCO3 export anomaly of a similar magnitude, in absolute terms, to what is observed for UKESM1-0-LL, GFDL-CM4, and GFDL-ESM4 in CMIP6 (see Fig. 2a). The text has been corrected accordingly to consider this constraint on CaCO3 production as an independent coefficient from the production ratio parameter.
"In both simulations, the PIC production in the model was multiplied by $(1-\alpha_{carb})$ for carb– and $(1+\alpha_{carb})$ for carb+, with $\alpha_{carb} =1/3 \cdot (CO2-285)/(936-285)$"

145 – add ", respectively" after "sDIC"
→ This has been done.

146-147 – The authors should specify what tolerance for "drift" was used to exclude these models but include others with less drift.
→ This has now been more precisely specified:
"An sAlk drift threshold of 2 mmol m$^{-3}$ per century was used to exclude models when reporting ESM values that may be affected by such a drift. This criterion thus excludes CMCC-ESM2 and CNRM-ESM2-1, due to a salinity drift for the former and an Alk drift for the latter (see Appendix A5)."

198 – The "effect" should be more specific – do the authors mean chemistry "acidification", or fluxes "carbon uptake" or both?
→ Correct. "effect" has been changed to "acidification" here:
"Indeed, a decrease in PIC production would lead to a relative basification of the surface ocean, and thus to a dampening of the acidification related to the increase in atmospheric CO2 concentration, i.e. a negative feedback."

Figure 2 – yellow line "crab_low" should be "carb_low"
→ This has been corrected, also for Fig. B4.

249 – "that" should be "those"
→ This has been corrected.

250 – I think "affected" should be "driven"
→ This has been corrected.

Figure 3 – Why no symbols for NEMO-PISCES simulations on panel A? Unclear why a few symbols in A and C have a black box around them, "7" in panel C and "yr" in axis labels are not showing up, and Y-axis legend for panel D also seems to have formatting issues in my pdf version.
→ No symbols are plotted for the NEMO-PISCES simulations in panel A because the PIC export anomalies were directly constrained based on atmospheric CO2 concentration for a consistent POC export anomaly value. Furthermore, this POC export anomaly does not correspond to that of CMIP6 ESMs, as the ocean dynamics in our simulations remain those of the pre-industrial era.
→ Clarification of the black pointer outline in the various panels has been provided, detailing the reasons and the relevant ESMs in each case:
"Linear regressions of NEMO-PISCES simulations (dashed grey lines) are calculated using the standard, carb– and carb+ simulations. The carb_low simulation is excluded (indicated with a black pointer outline) as its initial state differs from that of the other simulations. CMIP6 ensemble linear regressions (solid grey lines) are calculated excluding ESMs with a black pointer outline. In panel (a), UKESM1-0-LL, GFDL-CM4 and GFDL-ESM4 are excluded as they consider a saturation state dependency for PIC production, leading to substantial differences in PIC export anomalies compared to the other ESMs. In panel (b) CMCC-ESM2 and CNRM-ESM2-1 are excluded due to their surface sAlk drift (see Sect. 2.3) and UKESM1-0-LL is excluded because it omits the influence of the soft tissue pump on sAlk (see Appendix B1)."
→ The formatting issues are not apparent in our pdf version, this will be checked in subsequent proofs.

278 – remove second "is"
→ This has been corrected.

281 – This sentence does not make sense to me and should be reworked "Although a calcite saturation state threshold can robustly be pointed out, the shift itself should be reversible regarding this environmental control parameter, and should remain dependent to it."
282-284 – "If subsurface Ωcalc is back at higher values than the threshold, then the vertical PIC dissolution should shift back at depth, probably without hysteresis." I should be made clear that this is resented as a hypothesis untested in he present study and thus speculation. As such, the next sentence shouldn't begin with "Therefore" as if the previous assertion had been proven. Also, it is not clear whether the Chen study was evidence for or against a tipping point. This should be clarified.
→ The few sentences closing this paragraph have mostly been removed, or rewritten for clarity:

"The shift in the dissolution pattern is robustly identified based on a calcite saturation state threshold, and the timing of its surpassing in a high-emissions scenario varies slightly depending on the strength and anomaly of the carbonate pump as well as ocean circulation. However, this shift is certainly reversible. If the subsurface $\Omega_{calc}$ returns to values higher than the threshold, the vertical PIC dissolution should revert to deeper depths, likely without hysteresis (Chen et al., 2021)."

288 – add comma after "uptake"
→ This has been corrected.

293 – remove comma after "uptake"
→ This has been corrected.

316-317 "As there is a robust negative correlation between CaCO3 export anomalies and salinity-normalized surface alkalinity in the CMIP6 ensemble, salinity-normalized surface alkalinity observations could be used to identify historical trends in PIC export." Are the authors sure about that? Carter et al (https://agupubs.onlinelibrary.wiley.com/doi/10.1002/2015GB005308) argue that the earliest historical signals only emerge in 2030.
→ The original text was insufficiently clear. It has been revised to convey that these trends are expected to be observable in the future, likely within a decade or more, depending on the regions. Additionally, a reference to Carter et al. (2016) has been included.
"As a robust negative correlation exists between projected anomalies of CaCO3 export and salinity-normalized surface alkalinity in the CMIP6 ensemble, salinity-normalized surface alkalinity observations could be employed to identify trends in PIC export, with signals expected to emerge from time series data in the coming decade or so (Carter et al., 2016)."